# Quality Management Framework for Housing Construction in a Design-Build Project Delivery System: A BIM-UAV Approach

Amir Faraji [1,2], Maria Rashidi [2,*], Tahereh Meydani Haji Agha [3], Payam Rahnamayiezekavat [2] and Bijan Samali [2]

1. Construction Project Management Department, Faculty of Architecture, Khatam University, Tehran 1991633357, Iran; a.faraji@khatam.ac.ir or a.faraji@westernsydney.edu.au
2. School of Engineering, Design and Built Environment, Western Sydney University, Sydney 2747, Australia; p.zekavat@westernsydney.edu.au (P.R.); b.samali@westernsydney.edu.au (B.S.)
3. Construction Project Management Department, Khatam University, Tehran 1991633357, Iran; t_tahereh@yahoo.com
* Correspondence: m.rashidi@westernsydney.edu.au

**Abstract:** Quality management in project administration can affect the costs and schedule of a project considerably. The immediate notable result of unacceptable quality in a construction project is the "dissatisfied client." which can be interpreted as "customer loss". Additionally, defective work is a synonym for problems such as dispute, because items of non-compliance with early agreements can be considered the main factor in triggering claims by the client. The integrated use of two modern technologies, namely Building Information Modeling (BIM) and Unmanned Aerial Vehicles (UAV) is proposed in this study to support project quality management (PQM). This study aims to develop the theoretical underpinnings to provide a quality management framework, formed by BIM-UAV, for housing projects in design-build (D&B) contracts. For this purpose, in the first step the causes of client dissatisfaction rooted in quality concerns were identified in two phases of design and construction. The next step was dedicated to the mapping of BIM-UAV capacities to address the identified causes of dissatisfaction of the client. In the final step, expert opinion was obtained to integrate the BIM-UAV capacities to the quality management framework. The findings and main contribution of this study to the body of knowledge is a guide for design-builders to implement BIM-UAV as an innovative quality management solution to improve their services and to gain the maximum level of client satisfaction, focusing on house building.

**Keywords:** quality management; building information modeling (BIM); unmanned aerial vehicle (UAV); housing projects; design-build contract





## 1. Introduction

Over the past decades, a number of quality management alternatives have been developed and used in the construction industry. The word "quality" is derived from the Latin word "quail" meaning "what kind of". Defining quality is difficult or impossible because it has different types of meanings and implications attached to it, hence it is referred to as "slippery concept" [1]. According to the American Society of Quality, quality refers to the characteristics of a product or service that indicates its ability to meet expressed or implied requirements. The quality needs of project stakeholders are subjective and depend on their lifestyles, delectations, traditions, social structures, and education [2]. Quality management in construction achieves compliance with the need for all stages of construction agreed upon by the project stakeholders. This emphasizes the definition of quality policies that are implemented through planning, assurance and quality improvement. Quality is also defined as "conformance to recognized necessities," and quality management as "the worries optimization of the quality actions complicated in manufacturing a product, process, or service, with avoidance and evaluation actions" [3].

Project quality management (PQM) considerably affects deliverables of construction projects [4], and many aspects of project management knowledge directly or indirectly relate to this important area [5–8]. Housing projects are the heartbeat of urban modernization and contribute to local economic and cultural development [9]. In the global marketplace, increased levels of competition have resulted in the growing importance of quality for organizations, and, as a result, PQM has become a key organizational performance issue [10]. Generally, at the end of the design phase, quality concerns are considered by the employer because the major decisions about the construction phase should by then have been made be made by the client [11]. The employer's reaction at this stage shows the level of satisfaction and, in the case the design is not of an acceptable quality level, the dissatisfaction of the client triggered conflicts. The quality issue is discussed in the next section at the implementation and design phase. Therefore, the problem of quality should continuously be addressed within different phases of project for the satisfaction of the customer [12]. The following questions should be considered. Have standards, rules and governing regulations been observed in the project? Have materials met required standards? Are the sources of material supply approved by the employer? Is the employer completely aware of the time, cost and risks of the selected materials? Are the methods used in accordance with the governing rules and, among the various methods approved by the standard, does the employer agree with the selected method or not? There are many other related questions.

There is no integrated flow of information associated with the process of identification, reporting and rectification of defective building works. Secondly, an efficient mechanism to update project documents about the location of the defective works and the progress of corresponding remedial works has not yet to be established. In recent decades, emerging information technologies such as cyber-physical systems, the internet of things, augmented reality, point cloud, and Blockchain, are promising developments in product service systems [4,12–16]. The rapid development of these technologies has greatly enhanced the total volume of building sector data they can provide.

It is an accepted norm that the client defines quality in housing construction [17]. Therefore, exploration of novel technologies as quality management tools must be accompanied by the knowledge of factors that construe client satisfaction. The matching of BIM-UAV features capable of enriching information required for addressing the root causes of client dissatisfaction is the backbone of the proposed quality management framework in this research. Due to typical planning and execution of work, the majority of housing projects can be defined in the design-build project delivery method, and assessment of quality should be followed and analyzed in a continuum of design and construction. Therefore, the design-build system was considered as the focal point of the research. The framework assures that access to and more effective utilization of extra information reduces the likelihood of faulty design and construction. The following sections include a gap analysis, a literature review, the methodology of research, analysis of gathered data and the two proposed models.

## 2. Previous Works and Gap Analysis

### 2.1. The Importance of PQM Research

Previous studies have focused on project quality management that has investigated the quality area from different perspectives. The work in [9] emphasizes that urban construction must be quality-oriented, so the reason for sharing information by BIM is to enhance construction quality regarding standards. This paper was based on BIM technology that studied the quality control of the construction of complex urban projects using AR. Additionally, it is said that with the introduction of new technology and methods such as artificial intelligence in the construction process, and the use of more advanced methods and processes, the quality of project construction improves. In this model, the BIM can manage the manufacturing process and conditions, as well as model connections, and decrease problems between participants. In other research, the cooperation requirements

were analyzed, implementation technologies justified, and a process model developed for the cooperation of several stakeholders. This investigation proposes an approach that develops the construction quality management process with a BIM-based system and internal positioning [18].

Reviewing previous research shows that BIM-based technologies can provide positive solutions to quality problems and improve their performance over the life cycle of the project. Therefore, the quality and efficiency of labor resources, materials and equipment can be increased through a BIM-based system [19]. The use of BIM technology in the construction sector provides the potential for performing fast multiple operations, especially in the supervision works and later in the building management [20–22]. The BIM model is developed on the basis of multitask collaboration between all stakeholders and expertise that contributes to the project [23–25]. Despite this, as well as the benefits of implementing BIM in the construction industry, its acceptance in companies and project offices faces significant organizational challenges [26]. In manufacturing quality control projects, few studies have been conducted on the use of BIM and data management. The purpose of implementing BIM in quality management is to use its capacity to provide multidimensional data, combining design data and time sequences in quality control processes [27]. Every construction project is unique, and quality is always changing from time to time and place to place [28]. To better construct the full data interrelationships for quality management purposes in BIM, the data from BIM should be carefully organized and enriched. A product, organization, and process (POP) modeling method to complement 3D product models with process and organization models to support design and construction was proposed. The 4D program provides timely inspection and simulation of the entire process and helps project participants better understand the acceptance of quality and collaboration requirements. The paper argues that BIM is useful in eliminating conflicts and reducing rework in improving design quality, but little research has been done on its use during the project to control build quality and efficient use of information. It points out that BIM can be adapted to the current industry standard methods of quality management [29].

To provide a proactive approach to design quality management of building projects, another study focused on developing a design quality assessment tool based on the contributing causal factors and impacts of high-priority design defects. The developed implementation tool enables practitioners to assess project design quality and prevent undesirable risks in a timely manner. This study deepened the understanding of building design defects and supports design quality improvement through an effective design quality assessment tool. The study focused on providing a better method for design quality measurement on building projects. These findings provide a deepened understanding of the logic of design problems and represent a significant advance in the academic body of knowledge on design quality management on the building projects this study focuses on, creating a design quality assessment tool based on the study of defects and their effects on high-priority design to provide a preventive approach to the design of quality management of building projects. This research introduces a tool in Excel that designers and users can adopt to measure the quality of design with information in Word and answers to questions, risks and dangers to prevent the occurrence of defects [3]. The identification of CSFs that affect TQM implementation and examine the underlying factors for its implementation in the construction industry is the other major aspect of quality management. In this research, the two most important hidden factors influencing the quality of construction methods and the human factor, and their analysis is recognized. This article examines the key construction factors in the Brazilian construction industry [10]. In another paper, the identification of QM technologies for planning, reliability, control and improvement in highway construction was investigated. The identification of barriers and drivers of QM technology adoption was analyzed in this study [30]. Distributed security technology and pervasive information and communication technology were used to create new methods in intelligent construction. Blockchain and the Internet of Things can be used to promote secure data transfer and comprehensive quality monitoring. In this regard, monitoring

the time, cost and safety in prefabricated construction enables sustainable innovation with supply chain management [16]. In another attempt, the identification of important factors that have the greatest impact on the investment process of the residential construction sector in Poland was examined. These aspects include the performance of companies in the construction market, pro-government social policies, highly advanced technologies, and the use of correct market relations. This study investigated the importance of quality in housing projects [31].

Defects in the building interior, especially those that occur in the building structure, lead to a mismatch between the energy consumption predicted in the design and the operation of the building. Despite the quality management methods in social housing projects, defects affecting the thermal performance of housing are still an important issue. To increase the efficiency of quality goals with the aim of reducing energy consumption, changing the living standards of social housing tenants, and also reducing carbon emissions, a change in the quality management method is required more than before. The development and implementation of project quality programs has also been studied [32]. Development in the construction sector has been facing quality issues for several years and, therefore, the cost to the economy is great. If construction uses the concept of quality assurance, material damage is potentially reduced. The use of quality systems allows the company to organize its resources to achieve, maintain and improve quality economically. The need for quality systems in organizations is similar to the need for financial control systems, information technology systems and personnel management systems [33].

### 2.2. Gap Analysis in the Context of PQM-BIM-UAV

For the gap analysis step of the research, the main three concepts of study included project quality management (PQM), building information modeling (BIM) and Unmanned Aerial Vehicles (UAV), in paired combinations. Data from previous research showed that quality inspection from the viewpoint of the client using BIM and UAV has strong potential to model PQM in two different phases of design and build for housing construction. Table 1 summarizes previous related research in comparison to the current investigation.

**Table 1.** Comparative study of previous PQM-BIM-UAV research and the current investigation.

| Main Areas | Ref. | Year | Objective | Scope | Methodology/ Utilized Technologies | Results |
|---|---|---|---|---|---|---|
| PQM-BIM | [34] | 2014 | Investigating potential of using BIM for quality management in infrastructure projects | Highway and bridge construction | − BIM for quality management <br> − Quality assurance (QA) and quality control (QC) | Critical aspects of the utilization of BIM for quality management |
| | [29] | 2014 | Adapting BIM to the current industry standard methods of quality management | Construction project quality management | − BIM models in products and process | Model based on using and integrating information in model |
| | [9] | 2017 | Providing quality protection to ensure the application of construction quality standards | Construction of urban complex project quality control | − BIM technology <br> − Augmented Reality (AR) | Model based on BIM and AR |
| | [18] | 2018 | Proposing a software-based quality management system for more efficient result | Quality management programs in construction projects | − BIM <br> − Indoor positioning technology | Establishing a process model for the collaboration of multiple stakeholders |
| | [35] | 2018 | A platform for gathering, managing and controlling the quality of the management data | Vietnam construction project | − BIM-cloud (360) | A quality management model based on cloud computing, mobile devices, and the BIM |
| | [36] | 2019 | Developing a digital quality management system | Construction quality management in the rural areas | − Questionnaire to investigate the integration of 2D documents with 3D BIM models | List of capabilities of BIM to enhance the quality management |
| | [37] | 2019 | To investigate quality management using BIM | Execution phase of structural elements | − (BIM) <br> − augmented reality (AR) -web-based system | A model to enter the inspection data directly in a shared digital environment |

**Table 1.** *Cont.*

| Main Areas | Ref. | Year | Objective | Scope | Methodology/ Utilized Technologies | Results |
|---|---|---|---|---|---|---|
| BIM-UAV | [38] | 2017 | Evaluation of the performance of 4D BIM | Computer vision-based construction progress | − BIM <br> − Image based progress tracking and object detection techniques | UAV-based progress tracking systems to update the schedule and progress information |
| | [39] | 2018 | To diminish fatal, non-fatal, and property damage | Construction safety performance | − BIM in the design <br> − phase <br> − UAV in the construction phase | 4D/BIM-UAV-enabled safety management model based on IDEF0 language |
| | [40] | 2019 | Safety inspection procedure for better understanding detected risk issues | Water diversion construction projects | − BIM <br> − UAV | Safety inspection model, which consists of data collection, dynamic BIM construction, and UAV-BIM |
| | [41] | 2019 | To improve construction safety performance | Construction projects | − Literature review | Recognition of two approaches in the BIM-UAV |
| | [42] | 2019 | A model to preserve the identity of the respective civilizations | Historic buildings and cultural heritage | − 3D geometrics technologies <br> − UAV systems | Historic building information modelling (HBIM) model |
| | [43] | 2019 | Cultural heritage buildings were subjected to numerous maintenance interventions | Architecture of historic buildings | − BIM <br> − UAV <br> − Photogrammetry | A model to develop 3D reconstruction as-built drawings |
| | [44] | 2021 | Administration of payments and lien rights | Analysis of construction progress | − Blockchain-enabled smart contracts and robotic reality capture <br> − UAV <br> − UGV | An autonomous payment administration solution |
| | [45] | 2021 | To automate surface inspection | High-rise building | − UAV-based data collection for unreachable <br> − inspection areas <br> − BIM | An automatic inspection method of building surface for the inspection data collection, by integrating UAV and BIM |
| PQM-UAV | [46] | 2021 | To reduce the incidence of construction accidents and improve the safety performance of construction projects | Monitoring of unsafe behavior of construction workers in construction site | − Computer vision (CV) technology | Content-based analysis method to depict the historical explorations |
| | [47] | 2021 | Monitoring construction project | Construction projects | − UAV 3D photogrammetric | Real-time monitoring, with quantitative and qualitative methodologies |
| | [48] | 2021 | UAV integration in construction workplaces from a health and safety perspective | Safe co-operation of UAV with human construction workers | − VR visualization techniques <br> − UAVs | Safety challenges of UAVs |
| | [49] | 2022 | To bridge intelligent operation and maintenance | Bridge engineering, operation and maintenance | − BIM <br> − VR <br> − UAV | Integrated model for bridge operation management using BIM, VR, and UAV technologies |
| Project Quality Management | [10] | 2020 | Identify the CSFs that affect the implementation of TQM | Brazilian construction industry | − Examining key construction factors | Integration regarding the practitioners' perception in the light of 20 critical factors identified in the literature |
| | [30] | 2020 | Identification of QM technologies for planning, assurance, control, and improvement | Highway construction quality management | − Reviewing quality management methods | Suggesting a roadmap and formal process for evaluating the adoption of emerging QM technologies |
| | [16] | 2020 | Developing an intelligent platform based on service-oriented manners with practical case demonstration | Refabricated housing construction | − SPSS <br> − Blockchain <br> − Smart construction | Developing a platform-based smart product-service system of prefabricated construction |
| | [31] | 2020 | Identify a few pivotal factors with the greatest impact on investment processes management in the field of residential construction | Housing construction sector in Poland | − CFS in quality management | Identifying CSFs emphasized are related to specialists working on the preparation and implementation of projects |
| | [32] | 2020 | Managing defects affecting the thermal performance of dwellings | Quality management in UK social housing projects | − Social tenants' living standards <br> − Definition of quality management methods | Promote the achievement of quality objectives aiming to improve energy |
| | [33] | 2020 | Organizing and managing resources to achieve, sustain and improve quality economically | Project effectiveness in construction and management | − Quality management systems | Reviewing quality assurance and control and their integration in the construction sector |

**Table 1.** *Cont.*

| Main Areas | Ref. | Year | Objective | Scope | Methodology/ Utilized Technologies | Results |
|---|---|---|---|---|---|---|
| | [3] | 2021 | To provide a proactive approach to design quality management of building projects | Project design quality Deepening understanding of building design defects | − An Excel-based tool | Developing a design quality assessment tool based on the contributing causal factors and impacts of high-priority design defects |
| *The current study* | | | *Developing a Model to evaluate the client satisfaction in different phases of project* | *Housing construction using design-build project delivery* | − *Expert Judgment* − *BIM-UAV requirements* | *Developing a two-steps model based on the comprehensive check list to assess the client's satisfaction in the four different stages of design phase and the build phase* |

## 3. Literature Review of PQM in Design-Build Contract

### 3.1. Project Quality Management

Complexity of construction and customer needs for standards require the implementation of TQM. Some advantages with using quality management include decreasing quality costs, increasing employee job satisfaction, customer satisfaction and recognition, proper execution of work quality from start to finish, and worker safety [30]. PQM involves the processes of combining organizational quality attitude to create strategy and manage and devise project and product quality needs to achieve participant objectives. PQM attains buyer satisfaction through an emphasis on process development, client and provider meetings, cooperation and teaching [10]. Quality management contains all activities of the management tasks that define the quality rule, aims, duties and tools with resources such as quality scheduling, quality device, quality guarantee, and quality perfection, within the quality organism [50]. Construction product quality can be defined as the grade to which the specified or indirect requirements and the internal features are assured during construction [29,30]. Construction companies use multiple quality management systems such as TQM, JIT, and LSS, among many others for waste reduction and quality improvement, by eliminating defects in production. [51]. Based on the literature review of quality management in the design sector, three accepted design quality basics are fullness, accuracy, and suitability [3]. By studying various articles, it was found that meeting the needs and expectations of the client, together with reducing construction time and costs, are the most important definitions provided for QM in the construction industry.

### 3.1.1. Quality Management in the Design Phase

Many previous studies state that a lack in design and certification are among the most serious causes for delays and cost increases [52]. Many problems associated with increased time and costs on construction projects are caused by errors or insufficiencies in documentation. Inadequacies result directly in plan postponements or repetitions, and indirectly in augmented team jobs and, therefore, unsuccessful presentation [33]. Quality of design in the construction industry has long been an issue because it is vital to project success. Nevertheless, producing design proofs without any defects still poses a main challenge. The growing complication of design costs intensifies design problems, particularly for building projects. To ease increasing anxiety about deficiencies in design and certification, several tools to judge design quality and define design defects have been developed. These tools have design quality scales, but absent is a practical method to qualify or avoid design weaknesses during design processes. Design quality management involves anxieties about optimization of the quality actions involved in manufacturing a product, process, or service, including avoidance and assessment events [53]. If design and documentation are unfinished, differing, or wrong, they are considered deficient [3]. To achieve quality at project completion, the client must specify needs at each step, the designer has to use up-to-date specifications and good-quality material, and the contractor has to use efficient staff to meet the specified requirements [2]. When items are selected on the basis of small design bills, the quality of projects is often imperfect, which usually results added

costs to the employer [3]. Because design requires both objective and subjective features, measuring its quality is hard [54]. Past investigators have established and presented some valuation tools for use in design quality management. Unfortunately, contractors fail to recognize that non-conformance with required quality is more costly than the achievement of required quality [2]. As a result, design defects are one of the factors in the failure of project quality management. Therefore, the introduction of tools to address this problem in the design phase is one of the objectives of this research. The use of emerging techniques, such as BIM, introduces new tools. Customer dissatisfaction, and increased time and cost are some of the most important problems that result from improper and incomplete design. Considering this important shortcoming in the field of poor quality, and also the notable positive features of BIM, use of this tool in project quality management is vital and fundamental.

### 3.1.2. Reasons for Client Dissatisfaction

In construction, the product constitutes the completed facility, and the client is the final user of the facility. A construction project is recognized as effective when it is on time, economical and meet the conditions for the client's approval [1]. Lack of supervision during construction, and deficiencies of building materials, can cause poorer building quality. The achievement of a housing project is determined by the approval of three parties: the architect, contractor and customer [55]. Assessing the level of customer satisfaction in this industry is a means of measuring product quality, modifying shortcomings to growth competitiveness and achieving customer expectations. Thus, a scientific method is required to forecast predictable results and decrease risk [56]. Housing progress is carried out in two important steps: the acquisition step and the production step [57]. A defect is defined as "a deviation of a harshness enough to need corrective action," and a deviation is defined as an "exit from recognized wants." Such design mistakes and errors significantly lead to undesirable results, including cost overruns, timing postponements, requests for information, and construction rework [3]. Each of these can cause dissatisfaction because they increase the cost and time allotted for the design and construction of the project, and also lead to rework and destruction of the existing standard. The client expects that having requested a design, there will be specific expectations and features in the final plan and design, and observed rules and regulations. Not paying attention to these expectations is the first step in creating dissatisfaction for. Failure to comply with this issue in the long run will cause the client to ask for change and re-work, which will certainly increase time and cost, reduce the quality of the final product, and increase dissatisfaction. During implementation, non-compliance with standards and safety and health on the project site, and the use of substandard materials and non-standard methods, are important reasons for dissatisfaction of the client. To ensure satisfaction of the client, which is the main aim of quality management, it is necessary to know the reasons for dissatisfaction and to prevent their occurrence during design and construction of the project. In Table 2, the reasons for client dissatisfaction in the design and implementation steps of housing construction projects are extracted from various studied articles.

**Table 2.** Reasons of clients' dissatisfaction in housing project design and build contracts.

| Reasons of Dissatisfactions in Design Phase | Impact of Reasons in Life Cycle of Project | Ref. |
|---|---|---|
| Inappropriate procurement system (type of contract). Lack of progress in project,. Failure to implement the standard. Lack of safety and health on site of project. Use of inappropriate materials, delay in chain supply. Choosing inappropriate resources. Lack of financial stability. | Termination of cooperation and non-continuation of the project. Cost increase. Delay in project. Rework. Inappropriate integration | [1] |
| Lack of appropriate design. Defects in design documents. Lack of integration between architect and civil engineer and mechanical and electrical engineer. Lack of standards in designing. Lack of site identification by designer, design documents deficiencies. Lack of review and correlation of all the information available in all steps and preparation of design documents. Insufficient coordination between design disciplines. Lack of familiarity of the designer with construction materials and execution techniques. Simultaneous allocation of staff to more than one project. False and contradictory information from the designer. Copy of previous work. Postponing the solution of design problems to the construction steps. Lack of integration along supply chain linking service providers and between project phases. | Cost increase and delay in time of project. Change request in design. Delay in project. Rework | [58] |
| Construction methods and tools. | Impact of reasons in life cycle of project | [11] |
| Columns, beams and ceiling conditions. Paint and rooftop conditions. Cracks in walls, columns and beams. Lack of safety on site. Lack of supervision in the implementation steps. Good or bad condition of material used in building insulation. | Cost increase, Delay in project, Rework | [57] |
| Roof details, Mechanical controls, Exterior wall systems, Installed specialty equipment, Interferences of project works; | Cost increase. Delay in project. Rework. Changes in project process | [3] |
| Lack of sustainability. Incrementally increasing of carbon emissions and worldwide energy usage. Lack of attention to environmental conservation and sustainable development. | Production of waste. Additional cost | [16] |
| Frequent reconstruction; | Cost increase. Delay in project. Rework | [30] |
| Poor performance. Product of additional waste. Failure to pay attention to the expectations of the client. | Time and cost overrun | [10] |
| Lack of sustainability. | Production of waste | [59,60] |

### 3.2. Housing Construction Projects and the Challenge of Monitoring

Delivering projects with desirable quality is a problem for construction because of the lack of data and studies to improve quality progress [10]. As industrialization of buildings is required to increase, much attention has been paid to quality problems in the construction industry. At the same time, the industry determines that industrial construction is acceptable, and that improvement of engineering quality is significant. This not only protects the production and life of people, but building is also the source of life, survival and development of the company, and has a great impact on progress. Today, the quality management of housing engineering and production technology is mature, and has formed a set of complete quality management theories. However, the construction is still in the early stages of industrialization, and industry engineers have little experience in quality management. A complete quality management system in the construction industry has not yet been formed [61,62]. Workers and manpower are the resources that measure the success rate of a construction project [63]. The key to success in any construction project is identifying and understanding its effectiveness and quality factors [31]. Therefore, the importance of the housing sector in the construction industry makes the issue of quality management in this sector more important than in other cases [64,65]. The improvement of people's housing conditions is one of the main tasks. Housing construction with attention to quality management has made it possible to offer better housing conditions. Experience in building and using houses built earlier, under standard designs, has shown the need for improvement in the variety of flats, which is somewhat limited, and the aesthetic standards of architecture. Therefore, paying attention to quality management in this sector, and planning for its success, is among the important principles of the construction sector [66].

On the other hand, the role of Design and Build companies is growing in the construction industry. In a Design and Build project (D&B Project), the contractor is responsible for both design and construction based on the client's project brief. Typically, D&B projects

are undertaken when a high degree of cost certainty is required at the time of contract award, and when improved constructability is required. The competencies of the client and the contractor play a vital role in making D&B projects successful. Researchers have emphasized the importance of the client's involvement during the initial project steps because as the work progresses, the contractor will have to depend on the client's brief, which often fails to provide end-user requirements clearly. In the pre-contract step, issues arise because of unclear project scope and subjectivity of the assessments made in selecting the contractor's design. Team participation, knowledge acquisition, effective management actions, leadership, proper communication channels, good inter-party relationships, and high employee involvement contribute to project success. These factors are closely associated with TQM elements, and identifying how TQM elements can contribute to an effective approach in construction is therefore required. However, clients ignore D&B projects when project quality is more important to them than fast project delivery. Generally, D&B project clients are unhappy with their contractors' service quality. Thus D&B contractors have to adopt innovative management approaches to ensure project quality. D&B projects should be properly managed from the design and construction interface management to enable the relevant parties to acquire a deep understanding of project complexity and realize the need to optimize the quality and constructability of the designs. Quality-related roles and responsibilities in D&B projects are different from those of traditional projects. To improve the output quality of the D& B projects, researchers suggest assigning proactive quality roles to the D&B team. The present-day quality management systems are loosely structured, and the contractors are not motivated to improve project quality, which may apply to D&B projects as well [2].

### 3.3. BIM-UAV and Project Quality Management (PQM)

BIM is now the manifestation of the digital revolution within the construction industry and is the key enabler for increasing teamwork, data sharing and manufacturing [62]. In the real-time visualization of project growth, when an operator connects a floor in the BIM model, he or she can gain full data about its mechanisms, containing design, position, plan, and costs [67]. Once goods and services are provided as a package, digital services are more able to provide the necessities of owners compared to fully employing physical goods [16]. BIM is most often used as a means for visualizing and harmonizing AEC work, avoiding mistakes, refining efficiency, and assistant scheduling, and for safety, cost and quality management on construction projects [68,69]. BIM can also create and save data formed during the entire life cycle of a building project and can be applied to various fields [70]. Outputs of BIM represent a project management system to maintain the cooperation of distributed project teams in complicated projects by creating a board for operational data sharing with limited restrictions on calculating devices [18,29,71]. Therefore, predicting the organized trend of a building information modeling system, which has entered the construction industry in recent years, despite being nascent, can eliminate many of the problems and difficulties facing building design systems at the senior management level [72]. BIM offers a complete catalog that can be used not only for imagining construction products using 3D models but also for leading numerous investigations into the models [18]. Because of the stability of design information with quality data and construction process with quality control process, the potential of BIM application in quality management lies in its ability to present multi-dimensional data containing design data and time sequence [29]. The combination of new and existing technologies is expected to resolve quality management problems [18]. For instance, quality faults can be assessed with a BIM model by connecting the checked results of surface quality with the surfaces in the BIM model [4]. Housing construction should be quality-oriented, because of which BIM information sharing, features and functions for construction of the project offer quality protection to confirm construction quality standards [9].

The first application of BIM in QC is updating information and different steps of the project life cycle to reflect the role and cooperation of stakeholders, which facilitates

communication and information sharing between stakeholders in the project [9]. One of the main functions of BIM is to analyze constructability through simulation and increase productivity and reduce interference in construction projects and the cooperation of actors involved in the project at different steps of its life cycle [29]. Collaboration between the various roles involved in the project will ultimately reduce differences and lead to faster conclusions for important project decisions, such as the choice of materials and the type of construction method [73]. Changes in construction projects are among the inevitable factors. If these are not considered properly, they lead to a great deal of adverse conditions and have a negative impact on the project. BIM offers significant benefits in coordinating changes to a model. BIM is a revolutionary advance that has accelerated the transformation of architecture, engineering, and the construction industry [74–77].

Unmanned aerial vehicles (UAVs)-based data collection using light or thermal cameras, especially for inaccessible inspection areas, is the basis for unmanned inspection of building [45]. Using these technologies, three-dimensional models can be produced that allow different geometric configurations of the distribution of building elements analyzed using drones, 3D mesh creation, augmentation of environmental realities, and BIM. The use of new technologies to control project work increases accuracy, diminishes errors and saves time, resulting in significant cost savings and avoiding project deviation [78]. The flight plan, mission and the results of obtaining information are important factors in the management framework of drone utilization. UAV-based technologies such as Photogrammetry/ BIM/ AR-VR are ways to reduce the risk of data inconsistencies, monitor communications, and increase the impact of practical guidance on decision-making in building inspection activities. These technologies can generate informational and visual data that allow various stakeholders to be in the flow of the building inspection and monitoring [79]. Accordingly, two main subjects can be distinguished here: obtaining 2D and 3D data with acceptable quality and precision, and the availability of integrated systems in the design and build phases [78]. There are three different types of application for UAVs: as an instrument of measurement and representation, as a tool for the architectural design phase, and as a tool for reconsideration [78].

## 4. Research Methodology

To achieve our research objective, a review of the literature on the topic was conducted and related articles were reviewed. As a result of these studies and familiarity with definitions related to the subject of investigation, a list of reasons for customer dissatisfaction in the construction sector and design and implementation phases was obtained. The reasons for dissatisfaction were examined because these are the most important factor in quality management in the construction phase. Definitions were studied in previous research papers and it was found that the main concept in the definition of quality management is satisfaction of stakeholders, which, in this article, is examined with a focus on housing construction. Therefore, to obtain satisfaction, it is necessary to know the causes of dissatisfaction, which were extracted from the study of articles and are presented in the Table 2.

These items were then used to develop questionnaires with twenty questions to assess their importance in terms of scoring. In this regard, experts were selected with the following criteria to develop questionnaires:

- At least a Bachelor's degree in architecture and civil engineering and a Master's or PhD degree in project management.
- More than twenty years of practical experience in the field of construction projects (employment in construction sites with the position of project manager and site supervisor).
- Employment in construction contractor companies with rank 1, as a senior with more than 10 years of experience, and employment in implementation and planning for implementation.
- Employment in consulting companies with rank 1 as a senior with more than 10 years of experience and employment in the design and planning department.

According to the criteria set among companies and individuals, 73 were recognized. Of these 73, 16 had a history of about 30 years building housing projects, 24 people worked in first-class consulting companies with more than 20 years of experience and in the design department as the head of the design office, 17 people worked as contractors in project management and planning, and 16 had PhD degrees and taught at universities and other academic institutions. Questionnaires were provided to these people, and they expressed and commented on the reasons for the dissatisfaction of clients and customers in design and implementation projects. After reviewing the answers, face-to-face conversations, and interviewing these people, checklist questions were designed. Answering these questions, which should be done by clients and customers, will have a direct impact on the quality management of a project. Questions in two sections related to the design phase and the build phase were extracted from the answers of the experts. The scoring of the questions was as follows. Experts were asked to rate the questions in five groups. Questions with a score of 0 to 20 are unnecessary questions and do not need to be on the checklist. Questions with a score of 20 to 40 are questions that can be affected to a small extent in measuring the satisfaction of the client. Scores of 40 to 60 are related to questions whose presence and answers above 50% are effective in measuring the satisfaction of the client, and questions that score 60 to 80 are considered necessary questions that should be re-examined and certainly be on the checklist. Finally, questions that score 80 to 100 are very important questions that must be on the checklist. In the design section, the opinions of experts were that the different phases of design should be separated and based on their opinion. Through two rounds, four sections were considered for the design phase, as mentioned in the next section. Accordingly, 134 questions were asked for the design phase and were provided to the experts to rate the questions. Of the 134 questions asked, 39 were in the 80 to 100 group and were placed directly on the checklist. Twelve questions were in the 60 to 80 group and were reviewed and then placed on the checklist. Twenty-five questions ranged from 40 to 60, which were reviewed and reduced to six questions and placed on a checklist. Seventeen questions were in the range of 20 to 40, and were reviewed again and reduced to three questions included in the checklist. Forty-one questions were in the range of 0 to 20, and were removed from the questions due to being unnecessary. Finally, a checklist with 60 questions was prepared for the design phase in four steps, as used in model number 1. In the build phase, 205 questions were extracted, 95 of which were in the range of 0 to 20, whose answers, according to experts, had no effect on customer satisfaction, and were removed. Of the remaining 110 questions, 23 were in the range of 80 to 100 and 22 were in the range of 60 to 80. The first group was unchanged, and the second group was included in the checklist after changes. Thirty-nine questions ranged from 40 to 60, and were reduced to three questions after review. Twenty-six questions were in the range of 20 to 40, were reviewed and reduced to two questions. Finally, a checklist with 50 questions was prepared for the implementation phase in two parts: first, the procurement and logistics phase, and second, the operational and executive considerations phase, as used in model number 2. The methodology of research is illustrated in Figure 1.

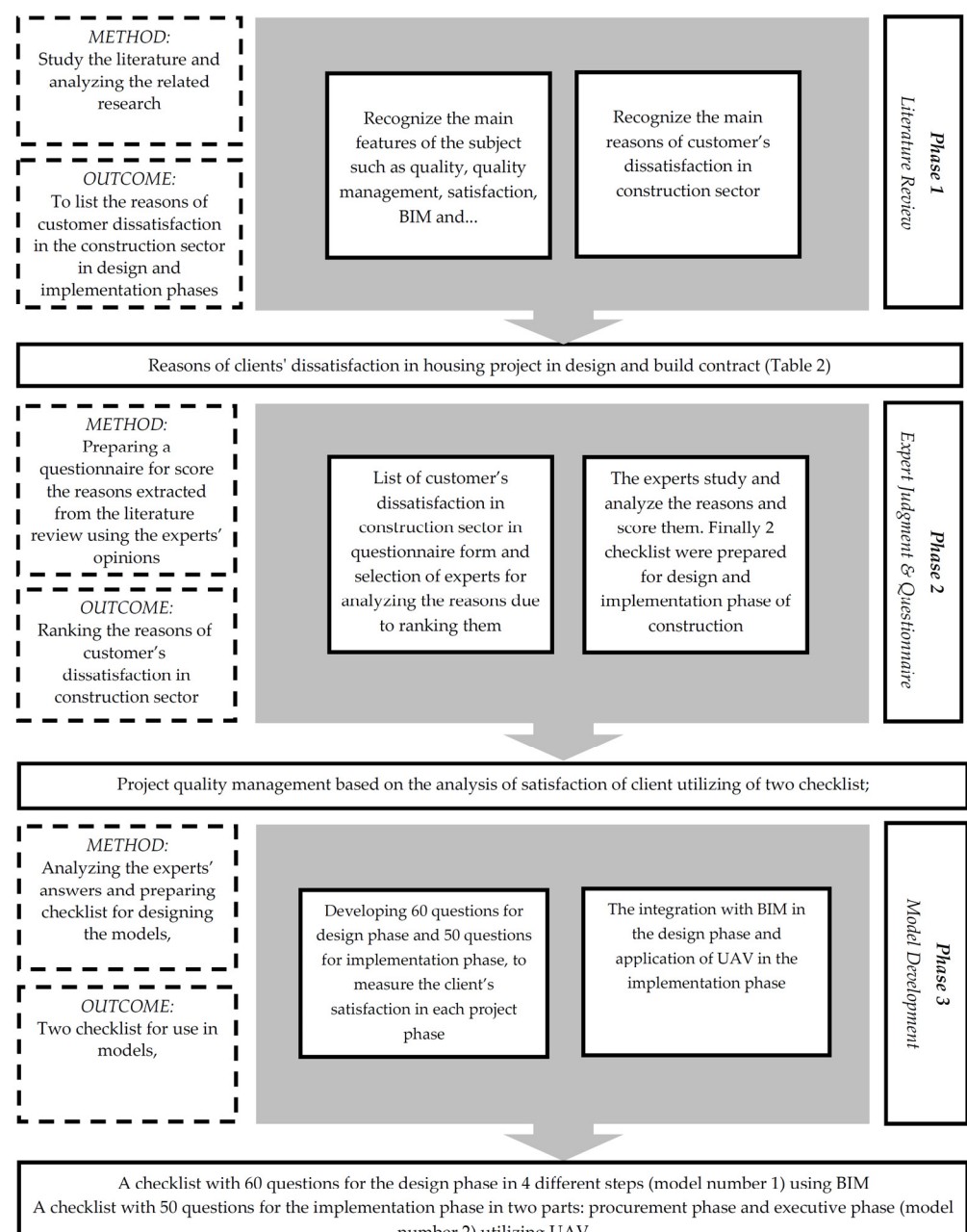

**Figure 1.** Overview of different phases of research.

## 5. Data Analysis

### 5.1. Stages of Design Phase

Based on an analysis of the experts' opinion on a breakdown of steps in the design phase of the housing construction sector, the different steps of design phase in housing projects were summarized as the following four stages (Figure 2):

- Conceptual design
- Basic design
- FEED design
- Detailed design

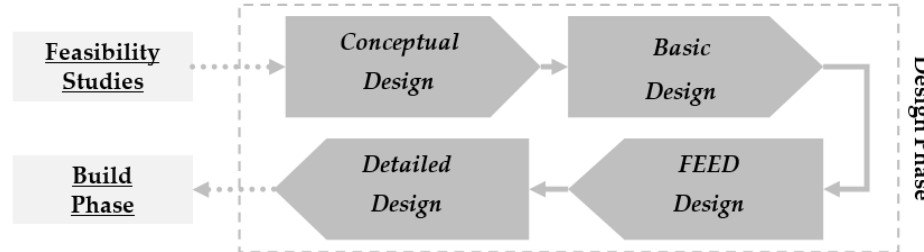

**Figure 2.** The four stages of housing design phase.

### 5.1.1. Conceptual Design

The conceptual design is the first stage of the project design process after the feasibility study phase. In the conceptual design stage, based on the defined requirements and solutions received as feedback from key project stakeholders, including owners, suppliers of goods, equipment and services and technical knowledge, the overall appearance of the project is determined. Additionally, design principles for the basic and detailed stages are developed. At this stage, the main processes and functions of the major systems that make up the project are determined, and more than one option is proposed to meet the project goals [80]. Conceptual architectural design is a complex process that draws on past experience and creativity to generate new designs. This process should be considered as an exploration of requirements, as well as of possible solutions to meet those requirements. [81]. Form finding and aesthetic aspects have always been an integral part of conceptual design, although there has been no evidence of predefined procedural processes for this stage [82]. The most important factors that should be considered in the concept design process are:

- Considering aesthetic principles and observing them;
- Considering technical and engineering issues for project implementation;
- Use of natural elements;
- Requirements of the client.

### 5.1.2. Basic Design

Basic design is a part of the design process during which the specifications of the main components of the design or project are determined based on a general outline of the design or project, as specified in the conceptual design stage, and by field study and engineering calculations. At this step, the processes are clearly defined and the main functions of all the systems that make the plan or project are identified. Analysis of process, equipment, mechanical and electrical systems, civil and structure systems and circulation of people is examined in this stage. The basic design stage is conducted based on initial data and by performing engineering calculations, as well as the specifications of the main components of the design phase. The designs presented in this section are:

- Process design;
- Equipment design;
- Piping design;
- Civil and structural design;
- Electrical design;
- Control and instrumentation design.
In the basic design, the philosophy and the basis of the design, the required criteria and standards, the necessary diagrams and drawings, and the preparation of the technical specifications of the systems are compiled separately in each section.

### 5.1.3. FEED Design

The term FEED stands for Front-End Engineering Design. It refers to a stage of the design process in a construction project that lies between basic design and detailed design. In FEED, the results of basic design stages of the project in different fields should come together and be combined before the detailed design begins. FEED experts, who are

generally experienced people, have an important role in project implementation because they estimate the financial cost of the project and provide acceptable technical and economic solutions. The FEED stage plays a major role in the project design because integration of different outputs of the previous design stage, including engineering specifications of materials, manuals for equipment s and guidelines of manufacturers about installation prerequisites must be taken into account.

### 5.1.4. Detailed Design

The detailed design stage is performed after the end of FEED, and in relation to work in parallel with it, to perform final calculations, prepare executive and construction plans, and to finalize technical specifications and other engineering documents. Detailed design is an important link between the design office and the project implementation phase. Based on the basic design information, in this stage, the final calculations, preparation of plans and construction, finalization of technical specifications and other documents are prepared. Most of the documents of this phase are plans and other executive drawings:

- Process: updating process data sheet forms, compiling control indicators;
- Equipment: technical specifications, determination of tests and technical inspection;
- Plumbing: preparation of models and drawings, isometric drawings, stress calculations and lists of valves;
- Precision tools: preparation of detailed maps, circuit diagrams, junction boxes and fire detection and alarm systems;
- Electricity: preparation of technical specifications of generators, transformers, control panels, ground connection and load calculations, connections and cables;
- Civil and Structures: preparation of plans of foundations, buildings, canals, sewage disposal systems and estimation of building costs;
- Details about interior design.

### 5.2. Measuring Client Satisfaction

In the next step of research, by studying papers, conducting interviews with experts and examining the reasons for customer dissatisfaction in the housing design and construction section, comprehensive checklists were developed. Table 3 shows a checklist to measure a client's satisfaction in the design phase, divided into four main stages. In addition, Table 4 shows the checklist developed to measure client satisfaction in the build phase. As discussed in the next section, these checklists are core elements of the proposed models to measure the clients' satisfaction as approval to passing each stage.

**Table 3.** The developed checklist to measure client's satisfaction in the four stages of design phase.

| | | Question of Checklist | Very Low 0–25% | Low 25–50% | Moderate 50% | High 50–75% | Very High 75–100% |
|---|---|---|---|---|---|---|---|
| Conceptual Design | 1. | How much does the design contract with you have a regular and legal framework? | | | | | |
| | 2. | How much do you agree with the proposed schedule for preparing all design documents (the length of time required to prepare all design documents)? | | | | | |
| | 3. | How much do you agree with the proposed cost of preparing the plan? | | | | | |
| | 4. | Do you trust the skills, experience and efficiency of the design team? | | | | | |
| | 5. | Are you satisfied with your relationship with the people in the design team? | | | | | |
| | 6. | Has the design team conducted feasibility studies to prepare the design? | | | | | |

**Table 3.** *Cont.*

| | Question of Checklist | Very Low 0–25% | Low 25–50% | Moderate 50% | High 50–75% | Very High 75–100% |
|---|---|---|---|---|---|---|
| | 7. In your opinion, how accurate are the climatic, cultural, social, artistic studies, aesthetics standards and criteria, etc.? | | | | | |
| | 8. In your opinion, how much has the design team studied the project site? | | | | | |
| | 9. Are the limitations and weaknesses of the site recognized by the design team? | | | | | |
| | 10. Is the design based on basic studies? | | | | | |
| | 11. Is the design based on the site conditions and related studies and in accordance with it? | | | | | |
| | 12. Are you satisfied with the number of options offered at the beginning of the design phase? (Has the design team offered you several options, and do you think that number is enough?) | | | | | |
| | 13. Is the site completely mapped and did the design team use the correct dimensions to design the project? | | | | | |
| | 14. Have neighborhood studies been conducted? | | | | | |
| | 15. Have different views of the site been captured by the design team? (Different views of landscape) | | | | | |
| | 16. Have access to the site been investigated and the movement of vehicles and pedestrians studied? | | | | | |
| | 17. Is the location of the building on the site correct | | | | | |
| | 18. Have you been asked "what do you want" before making a plan? | | | | | |
| | 19. Do you agree with the concept and form of the designed building? | | | | | |
| | 20. Do you agree with the concept and form of the designed facade? | | | | | |
| Basic Design | 21. Is the place suitable for different uses and function and furniture? (For example, appropriate place for kitchen, bath, etc.) | | | | | |
| | 22. Is the natural lighting of the spaces appropriate and sufficient? | | | | | |
| | 23. Is comfort of the spaces provided? | | | | | |
| | 24. Are public and private spaces in well-designed housing separated from each other? | | | | | |
| | 25. Is there inappropriate additional space in the proposed design? | | | | | |
| | 26. Has the designed structure disturbed the design? | | | | | |
| | 27. How much is the selected and proposed structure suitable for the building? | | | | | |
| | 28. Are the dimensions and area of the designed spaces appropriate and sufficient? | | | | | |
| | 29. Is the issue of sustainable development considered in the plan? | | | | | |
| | 30. Are all the requested spaces in the prepared plan? | | | | | |

**Table 3.** *Cont.*

| | | Question of Checklist | Very Low 0–25% | Low 25–50% | Moderate 50% | High 50–75% | Very High 75–100% |
|---|---|---|---|---|---|---|---|
| FEED Design | 31. | Do you agree with the proposed mechanical and electrical system for the building? | | | | | |
| | 32. | Has the designed mechanical and electrical system disrupted the architectural design? | | | | | |
| | 33. | How much do you agree on the amount of space occupied by installation ducts? | | | | | |
| | 34. | Do you consider the proposed materials suitable for taps, toilets and kitchens? | | | | | |
| | 35. | How much do you agree with the cost of implementing the proposed and designed mechanical and electrical system? | | | | | |
| Detailed design | 36. | Do you consider the building materials intended for the structure suitable? | | | | | |
| | 37. | Do you consider type of the selected building materials suitable for the construction of interior and exterior walls? | | | | | |
| | 38. | Do you consider type of the selected building materials suitable for doors and windows? | | | | | |
| | 39. | Do you consider the type of the selected building materials and their color suitable for flooring, wall covering and roof covering? | | | | | |
| | 40. | Are you satisfied with the interior design and furniture used? | | | | | |
| | 41. | Are the brand and technical specifications of the proposed and designed materials presented in the drawings? | | | | | |
| | 42. | In your opinion, is the detailed plan prepared complete and has all the necessary details to be implemented? | | | | | |
| | 43. | Did you see a 3D simulation before executing? | | | | | |
| | 44. | How well do you know the 3D quality provided? | | | | | |
| | 45. | Does the presented 3D show all the details? | | | | | |
| | 46. | Have all the rules and regulations in the proposed plan been observed? | | | | | |
| | 47. | Are the criteria related to urban landscape regulations considered in the presented facade plan? | | | | | |
| | 48. | Is there coordination between the facade design of your building and neighboring buildings? | | | | | |
| | 49. | How much do you agree with the coordination between the designed facade and the existing neighbors' facade? | | | | | |
| | 50. | Does the time required to execute the intended details match your schedule? | | | | | |
| After Design Is Prepared | 51. | Are your needs and expectations included in the final design? | | | | | |
| | 52. | In different steps of preparing the plan, have you observed it and provided your opinions about it? | | | | | |
| | 53. | Have your opinions been effective in preparing the plan? | | | | | |
| | 54. | In case of dissatisfaction with parts of the plan, have the necessary changes been made in those parts? | | | | | |
| | 55. | Has the designer regularly asked you to comment on the design? | | | | | |

**Table 3.** *Cont.*

| Question of Checklist | Very Low 0–25% | Low 25–50% | Moderate 50% | High 50–75% | Very High 75–100% |
|---|---|---|---|---|---|
| 56. In your opinion, how much has your level of satisfaction with the various steps of preparing the plan affected the final design? | | | | | |
| 57. After preparing the design, have your positive and negative comments been fully considered in making the design changes? | | | | | |
| 58. In your opinion, are the design documents complete? | | | | | |
| 59. How much has the design team paid attention to your requests and needs and applied your ideas and met your requests? | | | | | |
| 60. Is the cost of providing the proposed materials for housing construction commensurate with your budget? | | | | | |

**Table 4.** The developed checklist to measure client satisfaction in the build phase.

| | Question of Checklist | Very Low 0–25% | Low 25–50% | Moderate 50% | High 50–75% | Very High 75–100% |
|---|---|---|---|---|---|---|
| | 1. How satisfied are you with the legality of the execution contract? | | | | | |
| | 2. How satisfied are you with your needs and expectations in the contract? | | | | | |
| | 3. Is the level of preparation of executive documents complete with high availability? | | | | | |
| | 4. How much construction started at the announced time? | | | | | |
| | 5. How much necessary infrastructure such as water, electricity, etc. is properly provided for the construction site? | | | | | |
| | 6. How suitable is the site equipment for people that are in the place of work? | | | | | |
| | 7. What is the potential for planning appointments for periodic visits? | | | | | |
| Procurement and Logistic | 8. Is there appropriate space for holding necessary meetings? | | | | | |
| | 9. How good is the relationship between you and the executive team? | | | | | |
| | 10. How much does the executive team coordinate with you? | | | | | |
| | 11. How much material is procured from reputable and certified sources? | | | | | |
| | 12. How much materials are of good type and have a standard certificates? | | | | | |
| | 13. How much is the cost of procuring materials paid to the suppliers in a timely manner? | | | | | |
| | 14. How good is the relationship between the implementation team and the supplier of materials? | | | | | |
| | 15. Is the subcontractors' expertise adequate for the work assigned? | | | | | |
| | 16. How much of the implementation methods used are approved and standardized? | | | | | |
| | 17. Are the wages of the subcontractor and workers paid on time? | | | | | |

**Table 4.** *Cont.*

| | Question of Checklist | Very Low 0–25% | Low 25–50% | Moderate 50% | High 50–75% | Very High 75–100% |
|---|---|---|---|---|---|---|
| | 18. How well has the plan been implemented on the ground? | | | | | |
| | 19. How much excavation was done based on the elevation codes in the plan? | | | | | |
| | 20. Are you satisfied with the amount and results of concrete tests? | | | | | |
| | 21. Are you satisfied with the amount and results of welding tests? | | | | | |
| | 22. To what degree, when constructing a structure, the principles and rules of proper execution of structural elements such as foundations, columns, beams, shear and load-bearing walls, roofs and braces, etc. are observed correctly? | | | | | |
| | 23. How much are the principles of standard implementation observed during construction operations? (Principles related to the arrangement of walls, frames, retaining frames, etc.) | | | | | |
| | 24. How much has been verticality, alignment and absence of cracks in building elements such as columns, beams, ceilings, walls and etc. been controlled? | | | | | |
| | 25. To what degree are the principles of standard implementation observed during finishing operations? | | | | | |
| | 26. How suitable and correct are the insulations? | | | | | |
| Operational and Executive Considerations | 27. To what degree is the desirability of the type and color of the final interior coverings of walls, facades, floors, under ceilings? | | | | | |
| | 28. To what degree is the desirability of materials and color of the executed facade? | | | | | |
| | 29. How much are the implemented mechanical and electrical systems in the actual construction in the place embedded in the plans? | | | | | |
| | 30. How good is the equipment installed in the mechanical and electrical installations? | | | | | |
| | 31. How much does the interior decoration equipment match your aesthetic desires? | | | | | |
| | 32. How well does the implemented interior design match your requirements | | | | | |
| | 33. How many supervisors approved by the responsible organizations in the country were present at the site during the executive operation? (Engineering System Organization, etc.) | | | | | |
| | 34. To what degree did reviewing the technical checklists satisfy you? | | | | | |
| | 35. How satisfied are you with the guarantee and insurance of the insulations performed? | | | | | |
| | 36. How much do utility companies guarantee and insure their product? | | | | | |
| | 37. How much does the implementation team insure the constructed building? | | | | | |
| | 38. How much have appropriate enforcement methods been used to protect adjacent buildings and the walls of the excavation site? | | | | | |
| | 39. How much safety has been observed by workers? (Safety issues related to working at heights, welders, diggers, insulators, electrical workers, construction workers, etc.) | | | | | |

**Table 4.** *Cont.*

| Question of Checklist | Very Low 0–25% | Low 25–50% | Moderate 50% | High 50–75% | Very High 75–100% |
|---|---|---|---|---|---|
| 40. What is the level of monitoring the issues related to sustainable development (environment, no waste of energy, no waste generation, safety and health of people (HSEQ))? | | | | | |
| 41. How many steps of the project were completed in the estimated time? | | | | | |
| 42. How much was the cost incurred according to the cost plan? | | | | | |
| 43. How safe were the project site and people during the project? | | | | | |
| 44. How many accidents occurred during the project that resulted in casualties? | | | | | |
| 45. How many accidents occurred during the project that resulted in death? | | | | | |
| 46. How much work had to be repeated during the project implementation time? | | | | | |
| 47. To what extent was there a change in the sheets or a change in the execution process and predetermined programs during the project time? | | | | | |
| 48. How much was the construction of the building based on the prepared plan? | | | | | |
| 49. How much construction was provided based on the 3D model? | | | | | |
| 50. What was the impact of the checklist and model number 1 on the success of the project process in the implementation phase? | | | | | |

## 6. Findings and Discussion

### 6.1. Introducing BIM Model for Use in the Design Phase

Model number 1 was proposed according to the literature review and BIM potentials studies. Performing the design phase based on this model leads to the presentation of project results in a way that satisfies the client and thus increases the quality of the project because the final aim of quality management is to obtain maximum client satisfaction.

In one of the design stages, for example the conceptual phase, the model has three dimensions and is transferred to a plug-in in BIM using any design software. At this step, stakeholders, including the client, are requested to answer the checklist questions, otherwise the design team would be able to enter the next phase of the design and the next stage would not be unlocked by the system. The client uses the system, answers the checklist questions and assigns points, and as a result, the points convert to red, yellow or green outputs. This process can be done by other stakeholders defined in the system, and a result is obtained for this stage (as explained below) in order to continue. Scoring each stage can be done by one person or by several people on one panel. It is suggested that a construction start permit be issued if the result of the final checklist is recorded in green in the system. That is, the client is asked to answer the questions (checklists of Tables 2 and 3) and assigns points to each question. Finally, the analysis of the scores determines the activity status of the next stage, the points are calculated and three colors are obtained. Based on the experts' judgement, in the red color case (score range 0 to 30) the design is totally rejected. For the yellow color (range 30 to 70) the design should be edited, and the green color (range 70 to 100) means approval of the design and procedure to the next stage. The operation of model is defined as follow: The drawings obtain the necessary construction permits from the authorities when the client answers the checklist questions and signs with an electronic signature. The drawings are rated according to the three colors. The electronic confirmation of the client must also be included in the checklist of documents required for issuance of a construction permit by the authorities. Therefore, when referring

to a design-build company, the client must first prepare an electronic signature and a username and password. Then, at the end of each of the four design stages, they must log in to the checklist questions. After answering the questions, the scores are calculated in the main server system belonging to the project manager and the project manager is informed of the points earned at that step and the resulting color range. If the score is in the red range, the design is completely removed from the system and the design team must re-design the project. If the score is in the green range, it means that the plan is approved by the client and the next step in the software is unlocked. When defining checklist questions into the software code in a computer system, it must be specified which question is related to which design stage. If the score is in the yellow range, the necessary steps related to the questions that received a low score are unlocked and the user must edit them. If they are not edited, the software allows the user to go to the next steps; therefore, if there is a requirement that all stages be monitored by software controls, the user cannot use software to continue, and will not be able to control the next phases via the software. Finally, when the design and preparation of the necessary documents are completed, the client re-enters the system to score all the checklist questions. Then, the prepared design has a final score that may be in any of the three color ranges. It should be noted that trust in this system is assured when all users are required to enter all data in the software and also the client pays the cost of this project control model to the D&B company. A schematic view of the main elements and the major data flows according to the above proposed model is shown in Figure 3.

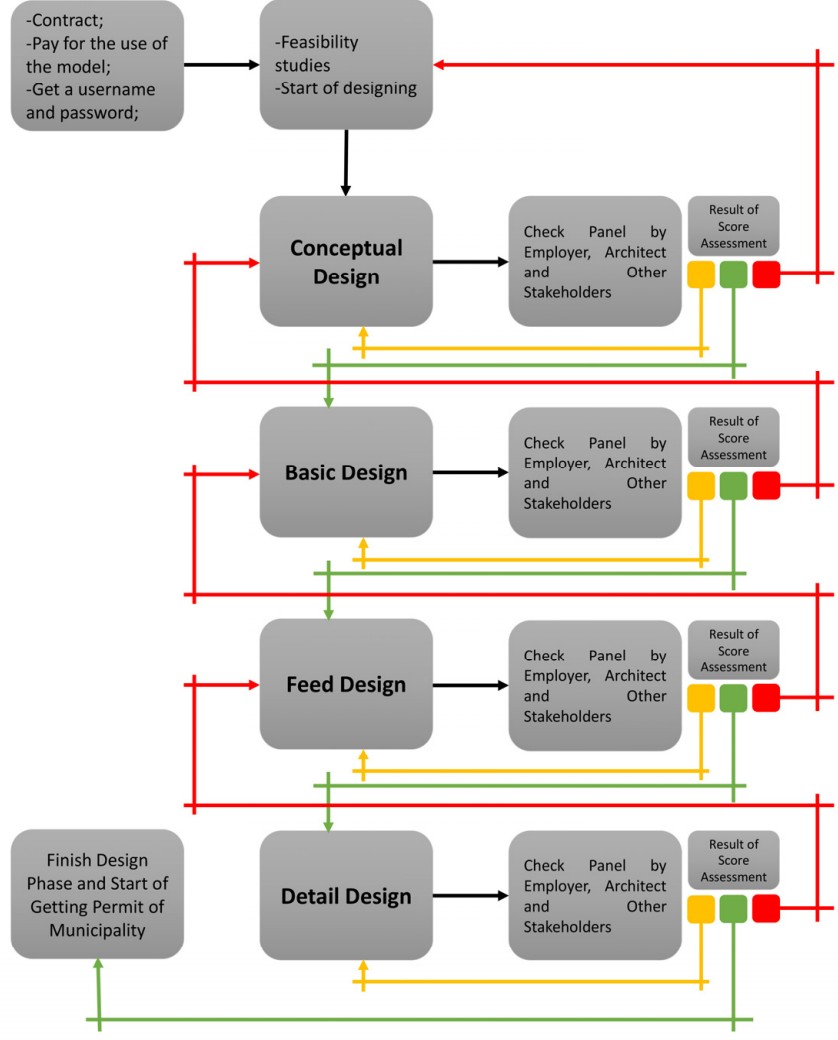

**Figure 3.** The BIM-based quality management model for the design phase.

### 6.2. Conceptualizing the BIM-UAV Model

As seen in the first model, if the design team obtains different views from the client during the design process and applies them to the project, a suitable design in accordance with the needs and expectations of the client will be prepared and sent to the authorities to obtain construction permits. It must be noted that the design should be in accordance with the needs and expectations of the client but should not contravene any rules and regulations and standards. In cases where the client's requests do not comply with the standards, the designing team should persuaded the client about the impossibility of the requested changes. To use the second model, the design must be prepared and approved, and the construction permits must be obtained from the authorities. In the build phase, it is suggested to use new tools such as drones and IoT sensors. The most important issue in this phase is appropriate use of the items mentioned. The construction and supervision team can determine what equipment to use at each stage, depending on the build phase considerations. For example, to maintain the safety of workers and other people, and to maintain the safety of the project site, it is appropriate to use a drone. Before starting, safety standards should be defined in the controller software. During construction, the drone takes photos of how to implement and maintain safety, and the prepared images are processed with the help of software systems. The processing results are transferred to the controller software and compared with standard data previously defined. In the case of deviation from the standard range, warnings of the work stoppage order are executed, and the human supervisor is obliged to stop the work. This model has several inputs, executable standards that should be converted to software-specific codes, acquired data from drones, standards and checklists.

The checklist questions are fed into the software during different steps of the work. Each checklist question has five points from the client viewpoint. The standard score has been defined already in the software model the score given by the client by observing (answering the checklist questions) and the score obtained by processing the items picked up by drones. The steps of work are under control in this model. For example, after one of the activities of the work breakdown structure (WBS) has been performed (foundation reinforcement has been completed). the task is checked and examined (e.g., using a drone) and the data obtained are processed and converted into points. Now, the data must be processed and measured. First, if standard measurements gain an acceptable score, based answers of the client to the checklist questions, the next processing step is started. The data are delivered to the client and, by logging into the system; the client accesses the checklist questions and gives them points. Now the scores are checked and the final score of the work is obtained, which is acceptable or unacceptable. If it is not acceptable, it enters the refinement step. The output of this model is a number that indicates the deviation degree of the output from the defined standard. If this number is within acceptable limits, the execution team will be allowed to continue working and start the next steps. If it is not within the acceptable range, there is a need to repeat work or repair problems until the obtained number is within the acceptable range.

As can be seen in the Table 3, the questions that should be answered indicate the level of satisfaction of the client with the performance of the work. In this model, in addition to described inputs, the client's answer to the questions are another input used to monitor the work performed. For example, in relation to maintaining the safety of the site, the client, by answering the questions related to this section, enters data into the software that is compared to defined standards. Therefore, the use of the checklist in Table 4 in this model is emphasized so that project implementation is based on the expectations and needs of the client. It should be noted that the implementation team must select the appropriate technology to be used in relation to each of the WBS activities in order to use it in a timely manner. The model must be defined in the system in such a way that confirmation of the correct performance of each step is dependent on obtaining an acceptable number of deviations from the comparison. By reviewing the scores given by the client, the processing score is prepared from the performance and compared with the defined standard to issue

a certificate of approval or disapproval of the performed step. A summary of the steps according to the above, proposed model as the second model for the build phase, is shown in the Figure 4.

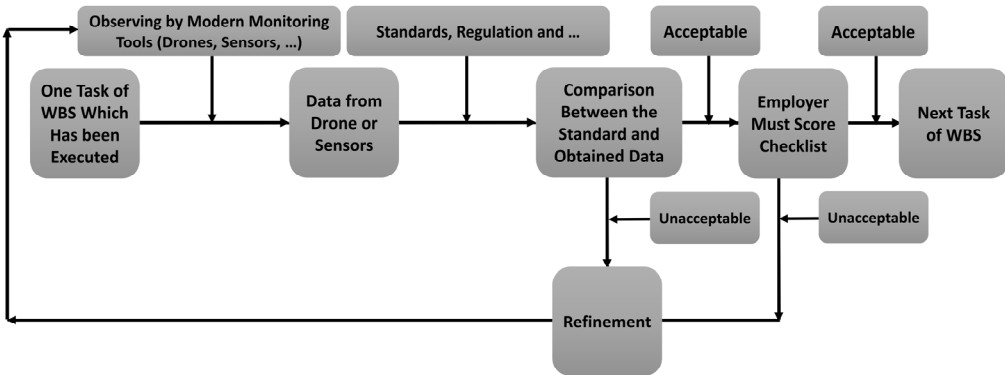

**Figure 4.** BIM-UAV based quality management model for the build phase.

*6.3. Practical Implications*

Project quality management in the construction context is related to the continuous measurement of customer satisfaction of implemented works, as well as conformity control with governing rules and codes. Although, monitoring and controlling the work in accordance with governing regulations is considered a common practice in the industry, satisfaction evaluation of the customer is still vague and challenging because such assessment depends on a vast spectrum of factors from the psychological make-up of the client to the behavioral skills of project manager. Since the continuous presence of corporations in the market is highly dependent on customers, it is essential for all businesses that have a service seller role in the construction industry to measure the client's satisfaction and to assure it. Nevertheless, the qualitative nature of the quality concept has stimulated decades of investigation into client satisfaction without a clear framework for satisfaction assessment. However, novel technologies and new tools and techniques make it possible to study this problem from new horizons and other perspectives. Nowadays BIM software techniques and UAV hardware technologies can improve the managerial procedures and processes. Based on the developed models, client satisfaction has been measured in two different steps, as describe below.

In the design phase, the client answers a questionnaire with 60 queries, at the end of each of four steps of design phase including conceptual design, basic design, FEED design and detailed design. According to the level of satisfaction of the client calculated using a five range scale, three different decisions can be made by converting them to a three-span scale. The three decisions, as shown for a given phase in Figure 5, are defined as "go ahead" to the next phase for a span 70 to 100, "need to refine" for 30 to 70 and "go to previous stage" when the score is 0 to 30. All the above process are defined in the model and the customer is engaged in the design process as a main stakeholder.

In the build phase, for data acquisition, a drone-based system can be used to monitor the implemented work and compare it with codes and regulations as the primary step. Meanwhile, the client is present, monitors the implemented activities/acquired data and reveals his/her degree of satisfaction by answering a 50 query questionnaire. As in the design phase and using five ranges and two span scales, the satisfaction of the client is measured and one of two options including "go to the next task" or "work refinement", as demonstrated in Figure 6, is selected.

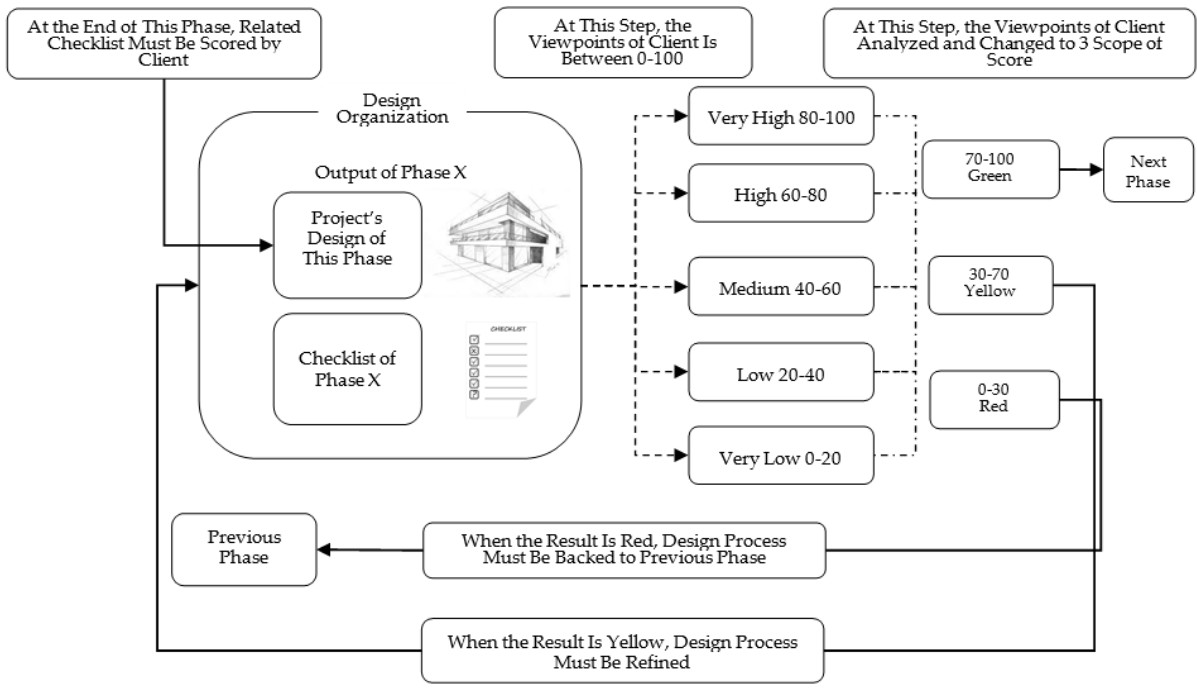

**Figure 5.** Flow diagram of the proposed model for the design phase of a construction project.

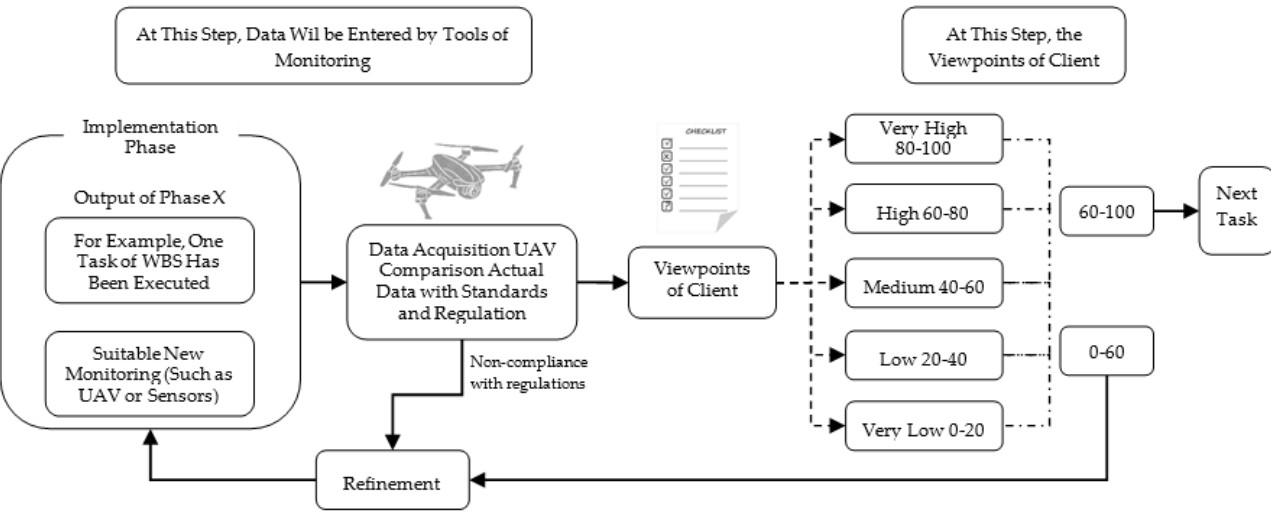

**Figure 6.** Flow diagram of the proposed model for the build phase of a construction project.

## 7. Conclusions

One of the important tasks in the quality management of construction projects is that the design, structure and materials used in construction should be inspected from time to time to ensure that the required standards are met and the expectation and needs of the client are satisfied. Monitoring is often done in a split manner; for example, by a civil supervisor, architectural supervisor, facility supervisor, and so on. This makes the process highly dependent on a specific supervisor in each discipline and may lead to errors. To enhance this process, new technologies such as BIM, IoT and a drone are proposed. Based on the proposed model, drone data are fed into software and at the same time, human supervisor and client comments are informed of the results. In addition, the expectations and needs of the client are entered as code in BIM software and the results of design and implementation are measured with these codes. In both the design and build phases, the described inputs, in addition to complying with standards, rules and regulations, are used

to measure the satisfaction level of client. For this purpose, two models are proposed: model number one for use in the design phase, and model number two for use in the build phase. It is worth mentioning that with use of these models in both phases, in addition to meeting standards, the expectations of the client are largely met and the most critical success factor of quality management is also met. Of course, the commitment of users to apply the proposed models and advance the work based on the steps mentioned is vital. Accordingly, key stakeholders of projects should commit to improve the process of quality management. Senior managers in contractor companies interested in developing a long-term relation with current and prospective clients, must expedite the process of quality measurement using reliable technologies. On the other hand, clients can insist and request tools and techniques which guarantee the desired quality throughout the project life cycle. Numerous elements during a project life cycle, such as complex processes, the external construction environment, and technical competence of practitioners, influence construction quality and restrict the effective practice of quality management. It is notable that results of the current research are limited by some parameters. For instance, since, arrangement of roles and responsibilities are entirely different in various project delivery systems, the model was developed in relation to design-build contracts. In addition, the model only considers housing projects in the vast spectrum of construction projects and, therefore, the questionnaire is limited to requirements of these types of projects. Developing ideas for applying novel technologies, such as UAVs as a means of quality detection, and assessment of client satisfaction using BIM, may have limitations in fulfilling their goals. However, with the rapid development of technology, the quality management of construction projects has gradually been transformed from empirical approaches into a technology-oriented perspective. It is hoped that in the near future, the use of new technologies will be mandatory in all construction projects and be among the requirements of project documents in the design and build phase.

**Author Contributions:** Data curation, A.F. and T.M.H.A.; Formal analysis, A.F., P.R. and T.M.H.A.; Investigation, A.F., M.R. and T.M.H.A.; Methodology, A.F., M.R. and B.S.; Project administration, A.F., M.R. and B.S.; Resources, P.R. and T.M.H.A.; Supervision, M.R. and B.S. All authors have read and agreed to the published version of the manuscript.

**Funding:** This research received no internal or external funding from any commercial or non-commercial source.

**Data Availability Statement:** All data, models, and code generated or used during the study appear in the submitted article.

**Conflicts of Interest:** The authors declare no conflict of interest.

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
