# Peer review of "Quality Management Framework for Housing Construction in a Design-Build Project Delivery System: A BIM-UAV Approach"

_buildings, doi:10.3390/buildings12050554_

Round 1

Reviewer 1 Report

Abstract:

The author also should include the impact or contribution of the study on the construction industry as well as related parties.

Content:

Overall is good.

In conclusion, my suggestion is the author also can tell about the limitation of study. The sentences also can be improved to clarify the impact or contribution of the study on the construction industry as well as related parties including clearer future research proposals.

Author Response

Dear Reviewer

We hope everything goes well with you

Many thanks for your valuable comments and precious time. Definitely your precise insights caused significant improvements to the text. However, what we done to enhance the investigation, can be expressed as below.

Sincerely

Main remarks of the reviewers and the related responses of the authors

No.

Reviewers’ Comment

Authors’ Description

1

The author also should include the impact or contribution of the study on the construction industry as well as related parties.

The abstract has been amended.

2

In conclusion, my suggestion is the author also can tell about the limitation of study. The sentences also can be improved to clarify the impact or contribution of the study on the construction industry as well as related parties including clearer future research proposals.

The conclusion section has been improved.

Reviewer 2 Report

This paper explores the applicability of BIM-UAV for Housing Construction in Design-Build Project Delivery System.

Overall the paper is well-organized and other than a few minor typos it has been written pretty well (i.e. In line 467 "housing" is spelled incorrectly.)

It is recommended that the authors elaborate on how they've chosen the criteria to select the experts to develop their questionnaire. 

Author Response

Dear Reviewer

We hope everything goes well with you

Many thanks for your valuable comments and precious time. Definitely your precise insights caused significant improvements to the text. However, what we done to enhance the investigation, can be expressed as below.

Sincerely

Main remarks of the reviewers and the related responses of the authors

No.

Reviewers’ Comment

Authors’ Description

1

Overall the paper is well-organized and other than a few minor typos it has been written pretty well (i.e. In line 467 "housing" is spelled incorrectly.)

The whole text has been reviewed.

2

It is recommended that the authors elaborate on how they've chosen the criteria to select the experts to develop their questionnaire.

The complementary descriptions have been added.

Reviewer 3 Report

This is a very good critical review paper, it is well written and has a great potential of helping stakeholders concerned with quality management of construction projects.

The reviewer appreciates the systematic approach adopted by the authors in the planning and preparation of the paper.

However, the following should be considered to further improve the paper:

  1. The title should read "Quality Management Framework for Housing Construction in Design-Build Project Delivery System: A BIM-UAV Approach" or "A Quality Management Framework for Housing Construction in Design-Build Project Delivery System". The methodology adopted may not be necessary in the topic
  2. the limitations of the study should be explained and justified. The choice of design-build over others should be explained while stating other procurement methods. Same for framework from client's perspective. Same for the BIM-UAV method.
  3. The comments in 2 can also be a source of identifying limitations of the study in the conclusion and recommendations. This will help in explaining areas for further study.
  4. There are some grammatical and syntax issues with the current paper, such as "......PQM has become a key organizational performance issue [10] including construction firms., etc. A proofreader should be engaged to fine tune the paper.
  5. I am not sure of the number of words but the paper looks too lengthy
  6. A major issue with this paper is the discussion of findings. The authors need to understand the difference between explanation of results and discussion of findings. what is available under the current section 6 is not different from explanation of findings
  7. For discussion, there is a need to compare the current findings with existing similar publications (while citing same) with a view to explain implications of the findings. More so, figures and tables should appear under findings and not discussion except where findings and discussions are done concurrently. 
  8. The conclusion should include limitations of the study, approach adopted to work around the limitations and areas for further study.
  9. Recommendation should be included as a section or as part of the conclusion. This should include policy recommendations for concerned stakeholders and how such stakeholders will will find the study useful.

Author Response

Dear Reviewer

We hope everything goes well with you

Many thanks for your valuable comments and precious time. Definitely your precise insights caused significant improvements to the text. However, what we done to enhance the investigation, can be expressed as below.

Sincerely

Main remarks of the reviewers and the related responses of the authors

Reviewers’ Comment

Authors’ Description

The title should read "Quality Management Framework for Housing Construction in Design-Build Project Delivery System: A BIM-UAV Approach" or "A Quality Management Framework for Housing Construction in Design-Build Project Delivery System". The methodology adopted may not be necessary in the topic

Thank you for your precious comment.

The title has been revised.

the limitations of the study should be explained and justified. The choice of design-build over others should be explained while stating other procurement methods. Same for framework from client's perspective. Same for the BIM-UAV method.

The complementary descriptions have been added to the text.

The comments in 2 can also be a source of identifying limitations of the study in the conclusion and recommendations. This will help in explaining areas for further study.

The complementary descriptions have been added.

There are some grammatical and syntax issues with the current paper, such as "......PQM has become a key organizational performance issue [10] including construction firms., etc. A proofreader should be engaged to fine tune the paper.

The text has been revised.

what is available under the current section 6 is not different from explanation of findings

Thank you for your precious comment.

The configuration of the subtitles has been amended.

More so, figures and tables should appear under findings and not discussion except where findings and discussions are done concurrently. 

The text has been revised.

The conclusion should include limitations of the study, approach adopted to work around the limitations and areas for further study.

The limitations of the research have been added to the text.

Recommendation should be included as a section or as part of the conclusion.

The recommendations have been added to the text.

This manuscript is a resubmission of an earlier submission. The following is a list of the peer review reports and author responses from that submission.

Round 1

Reviewer 1 Report

From the title "A Quality Management Framework Utilizing BIM-UAV for Housing Construction in Design-Build Project Delivery System", it was expected to reveal at least the two following components: a framework for quality management; a method to combine BIM and UAV utilization. However, the method in part 4 in insufficient to support both of the above.

1.Part 1:

1.1In line 77, "It is an accepted norm that the client defines the quality in the housing construction." As far as I know, the quality in the housing construction is evaluated comprehensively by the government regulator, the construction company, a thrid party for consultance and at last, the client.

1.2 Line 34-48, we have been all educated with the term "quality" years ago. Quality challenges in constrction are more informative and recommended in this part.

1.3 Line 79-81, "The matching of BIM-UAV features capable of enriching information required for addressing the root causes of client dissatisfaction is the backbone of the proposed quality management framework in this research.", unfortunately, I cannot find any other support of the "backbone" of the "matching of BIM-UAV features" in this article.

Part 2:

2.1 Even though long paragraphs are involved in the previous introduction to state the term "quality", the specific PQM is still unclear.

2.2 Line 91: I cannot find more supporting materials for "this paper is based on BIM technology that studies the quality control of construction of complex urban projects, using AR".

2.3 Line 127-138: which study? what approach? 

2.4 Line 139-159: I believe we all know the importance of quality in housing projects.

2.5 Line 169: "If the construction uses the concept of quality assurance, the material damage is potentially reduced." Invalid reasoning.

2.6 Line 175: Where is the GAP?

Part 3: Irrelevant contents.

Part 4: The literature review, the expert judgement & questionnaire and the model development are irrelevant to the topic of the title.

Part 5: Table 3 & 4: the checklists do not provide information about "BIM" and "UAV". Moreover, no data analysis or any results are covered in this part, I can only see questionnaires.

Part 6:

6.1 Line 550: a questionnaire is not a model.

6.2 Line 602: a questionnaire is not a model.

6.3 Fig. 3: There are no quality management actions in the figure.

6.4 The sum of the questionnaire score does not apply.

Part 7:

7.1: Line 712-716: no BIM, IOT or UAV are proposed.

7.2: This conlusion part is irrelevant to the whole previous parts.

Reviewer 2 Report

The manuscript presents the integrated use of two modern technologies (Building Information Modeling and Unmanned Aerial Vehicles) to support project quality management (PQM). In this paper is missing a deeper scientific discussion about the achieved results linked to indexed scientific works in this field. Discuss the results of your study with suitable references.

Furthermore it is necessary a better and clear explanation about the novelty of your study.

I strongly suggest to introduce a dedicated section devoted to the implications of the study that clearly shows what novel knowledge is created and what are the threats to the validity of the study. Which criticalities and which solutions? What are the future developments of the research work?

Reviewer 3 Report

1, Authors are recommended to proofread and polish the whole manuscript as a team before submitting it. For example, the first sentence in the introduction section is about quality management, while the following sentences are about quality. Then quality management and again quality. It's not easy to follow the logic. Also, in the second paragraph, housing projects are at the local level, but suddenly authors referred to the global marketplace. What's the relationship between them? What message exactly do authors want to send to readers? In lines 63-68, several questions are listed. Are the research questions or problems? I don't think so. Again, what's the authors' aim here? Please pay enough attention to academic writing. The entire introduction section is a puzzle for me, to be honest. Soon I lost interest in reading the following sections.

2, The flow issue of the storytelling appears here and there in the manuscript. In 2.1. The Importance of QPM Research, most contents are about BIM-based technologies, not what the title shows. Then I don't think I can trust other titles of sections and subsections. In line 96, what model? In line 133, what study? In line 139. What research? In line 145, what article? It seems to me that I know all words, but I don't know what I can get from these words.

3, Then suddenly, we came to Gap Analysis in the Context of QPM-BIM-UAV. Why UAV here? Little is mentioned previously.

4, I see the overlapping between 2. Previous Works and Gap Analysis, and 3. Literature Review of QPM in Design-Build Contract. Why not put all literature review and gap analysis together? It's quite strange to see 2.1. The Importance of QPM Research and 3.1. Project Quality Management in two separate sections.

5, Full names of abbreviations such as JIT and LSS are never given. By the way, I don't think JIT is a quality management system.

6, There is a lack of transparency in how data was collected. Which countries and regions? Which companies? Please provide their profile and tell us why they are representative.

7, I am unsure if housing projects and design and build contracts are necessary for this research. The links between these keywords are lacking. Why BIM-UAV is is the solution?

8, Please be noted that, for example, Stages of Design Phase is basic knowledge, not Data Analysis and Results. Without the empirical data, scholars and practitioners all know about this. In fact, no data is shown and analyzed in the section Data Analysis and Results.

9, following the above, The BIM-UAV based quality management model is proposed without empirical evidence and is superficial. Little contribution is found.

10, in summary, my impression is the fragmented sentences from different authors are randomly piled. Many parts of the article have nothing to do with the core, BIM-UAV, Housing projects, Design-build contract. I cannot really follow the article. Supervision is much needed. Please take responsibility!